# Equity Impact Assessment of Interventions to Promote Physical Activity among Older Adults: A Logic Model Framework

**DOI:** 10.3390/ijerph16030420

**Published:** 2019-02-01

**Authors:** Gesa Lehne, Claudia Voelcker-Rehage, Jochen Meyer, Karin Bammann, Dirk Gansefort, Tanja Brüchert, Gabriele Bolte

**Affiliations:** 1Department of Social Epidemiology, Institute of Public Health and Nursing Research, University of Bremen, 28359 Bremen, Germany; t.bruechert@uni-bremen.de (T.B.); gabriele.bolte@uni-bremen.de (G.B.); 2Health Sciences Bremen, University of Bremen, 28359 Bremen, Germany; bammann@uni-bremen.de; 3Institute of Human Movement Science and Health, Faculty of Behavioral and Social Sciences, Chemnitz University of Technology, 09126 Chemnitz, Germany; claudia.voelcker-rehage@hsw.tu-chemnitz.de; 4OFFIS—Institute for Information Technology, 26121 Oldenburg, Germany; meyer@offis.de; 5Working Group Epidemiology of Demographic Change, Institute of Public Health and Nursing Research, University of Bremen, 28359 Bremen, Germany; 6Leibniz Institute for Prevention Research and Epidemiology—BIPS, 28359 Bremen, Germany; gansefort@leibniz-bips.de

**Keywords:** equity impact assessment, logic models, framework, physical activity, older adults, interventions, intervention-generated inequalities

## Abstract

Reducing social inequalities in health and health determinants, including physical activity (PA), is a major challenge for public health. PA-promoting interventions are increasingly implemented. Little is known, however, about the impact of these interventions on social inequalities. For prioritizing interventions most likely to be effective in reducing inequalities, studies of PA interventions need to conduct equity impact assessments. The aim of this article is to describe the development of a logic model framework for equity impact assessments of interventions to promote PA. The framework was developed within the prevention research network AEQUIPA—Physical activity and health equity: primary prevention for healthy ageing, informed by an equity-focused systematic review, expert interviews, exploratory literature searches, and joint discussions within the network. The framework comprises a general equity-focused logic model to be adapted to specific interventions. The intervention-specific equity-focused logic models illustrate the key elements relevant for assessing social inequalities in study participation, compliance with and acceptance of interventions, as well as the efficacy of interventions. Future work within AEQUIPA will reveal which key elements are most critical for the interventions’ equity impacts. Equity impact assessments are beneficial for prioritizing interventions most likely to be effective in reducing health inequalities.

## 1. Introduction

Associated with various beneficial effects on health and wellbeing [1], physical activity (PA) is considered a major determinant of healthy ageing [2,3,4]. Epidemiological studies, however, indicate that the prevalence of being sufficiently physically active tends to decline with increasing age [5,6]. Moreover, PA has been shown to be unequally distributed across sociodemographic and socioeconomic population groups. Individuals of lower education, lower income, and ethnic minority populations tend to be less likely to be physically active [7,8,9,10]. Such social inequalities in PA are hypothesized to play an important role in explaining the social gradient in health, whereby a higher social position corresponds with better health [11,12].

Reducing differences in health between population groups related to socioeconomic or sociodemographic characteristics (i.e., social inequalities in health [13]) is a priority for public health. However, concerns have been raised about the risk of public health interventions unintentionally widening inequalities in health and health determinants between population groups, even where interventions are successful at improving outcomes across a population [14,15,16]. These ‘intervention-generated inequalities’ may arise at any stage of the intervention process, including during intervention provision, uptake, compliance, and outcome [17].

In 2007, Whitehead and colleagues [18] called for all public health interventions to be evaluated for their differential impact by socioeconomic and sociodemographic characteristics, including socioeconomic status (SES), gender, and ethnicity. This is often referred to as ‘health equity impact assessment’ [19]. The evaluation of equity impacts of PA interventions, however, is still scarce [20,21,22,23,24,25,26,27]. In the area of PA promotion for older adults, findings of our recent systematic review suggest that many studies evaluating the effects of interventions have not exploited the potential for assessing equity impacts so far [23]. When evaluating PA interventions, it is important to assess whether response to and effectiveness of the interventions differ by socioeconomic and sociodemographic characteristics [28]. However, practical guidance on how to systematically apply an ‘equity lens’ [29] to the evaluation of PA interventions is missing so far.

Given that all processes in the planning and implementation of interventions have the potential to affect inequalities [17], assessing equity impacts of interventions requires a comprehensive understanding of the intervention under study. In this regard, logic models of interventions, which can be defined as an approach to graphically illustrate and simplify the complex relationships between an intervention’s resources, activities, and outcomes, have been proposed as a valuable tool [30,31,32].

The aim of this study was to develop a logic model framework for equity impact assessments of interventions to promote PA among older adults.

## 2. Materials and Methods

### 2.1. Context

This study was carried out within the prevention research network “AEQUIPA—Physical activity and health equity: primary prevention for healthy ageing” [33] in the subproject “EQUAL—Equity impacts of interventions to increase physical activity”. AEQUIPA is funded by the German Federal Ministry of Education and Research (funding phase one: 02/2015–01/2018, funding phase two: 02/2018–01/2021). The prevention research network includes five subprojects (PROMOTE, TECHNOLOGY, OUTDOOR ACTIVE, RTC, AFOOT) focusing on the development, implementation, and evaluation of PA interventions for older adults using individual-, contextual-, policy-, or multilevel approaches (Table 1). Crucial for the integration of health equity aspects in the prevention research network AEQUIPA is the cross-cutting subproject EQUAL. During funding phase one, EQUAL aimed to synthesize the evidence on strategies to consider social inequalities in the development, implementation, and evaluation of PA interventions among older adults by means of an equity-focused systematic review [23] and expert interviews. Furthermore, EQUAL aimed at providing advice for the other subprojects on strategies to consider social inequalities, as well as developing a logic model framework for equity impact assessments of the five PA interventions implemented within AEQUIPA.

### 2.2. Development of the Logic Model Framework

Our equity-focused systematic review [23], qualitative semi-structured expert interviews, exploratory literature searches, as well as joint discussions within the prevention research network AEQUIPA informed the development of the logic model framework.

The equity-focused systematic review [23] was conducted to assess the extent to which and the ways in which studies on PA interventions for older adults consider social inequalities when evaluating intervention effects. The review was reported following the preferred reporting items for systematic reviews and meta-analyses-equity 2012 Extension (PRISMA-E) [53]. Eleven studies were identified that considered social inequalities when measuring intervention effects. These studies were examined to determine which methods to assess equity impacts were applied and whether the equity impact-related analyses were based on a theoretical framework or logic model.

Expert interviews were conducted to identify methods for considering social inequalities during intervention development, implementation, and evaluation. A convenience sample of twelve practitioners and researchers was identified through online searches, including searching the practice database of the German cooperation network ‘Equity in Health’ [54]. Based on experiences in social inequalities-sensitive intervention methods of health promotion among older adults as identified in descriptions of research focuses, conference talks, as well as published project reports and journal articles, seven researchers were selected as potential interviewees. Based on their experiences in PA promotion among older adults with particular attention to socially disadvantaged groups as identified in project descriptions included in the practice database ‘Equity in Health’, five practitioners were selected as potential interviewees. These twelve potential interviewees were invited to participate via e-mail. Interview guidelines were developed by the EQUAL research team aimed at filling in the knowledge gaps identified in our equity-focused systematic review as well as in view of the planned equity impact assessments. The guidelines covered the following topics: (1) Strategies for successfully recruiting diverse groups of older adults for taking part in interventions, (2) methods to consider sociodemographic or socioeconomic differences among participants during intervention development and implementation, as well as (3) experiences with equity impact assessments of interventions. The guidelines were modified in detail for use with the groups of researchers on the one hand and practitioners on the other hand (Appendix A). Five out of seven researchers and four out of five practitioners were interviewed. Interviews were conducted via telephone (*n* = 8) or face-to-face (*n* = 1), recorded, and transcribed verbatim. Main aspects of social inequalities-sensitive development, implementation, and evaluation were summarized by one author and discussed with a second author.

Exploratory literature searches were conducted to identify general literature on logic models, already existing logic models for equity impact assessments and PA interventions, and social inequalities-sensitive intervention methods for PA promotion in older adults. The literature searches were conducted by one author using PubMed MEDLINE and Google Scholar as well as examining reference lists of relevant articles. Search terms comprised keywords related to logic models, equity impact assessment, intervention-generated inequalities, physical activity, older adults, and interventions in various combinations.

Based on the findings from the systematic review, expert interviews, and literature searches, the EQUAL research team developed a first draft of the logic model framework, comprising a general equity-focused logic model to be adapted to specific interventions. Furthermore, the EQUAL research team developed first drafts of specific equity-focused logic models for the five interventions implemented in the AEQUIPA subprojects. In an iterative process of three rounds, all drafts were revised. In round one, the EQUAL research team discussed each draft of the intervention-specific logic models separately with at least one expert designated by the respective subproject. All discussion partners were scientific researchers experienced in the research area of the respective subproject and were required to be involved in the development, implementation, and/or evaluation of the intervention conducted in the subproject. Based on the results of the discussions, all drafts were revised by the EQUAL team. In round two, an equity-focused methodological workshop was conducted to jointly discuss the revised drafts of the general logic model and the intervention-specific logic models within the whole AEQUIPA prevention research network. Based on the results, the drafts were revised for the second time by EQUAL. In round three, each intervention-specific logic model was finalized by EQUAL in consultation with the responsible expert from iteration round one. Crucial for the whole consensus reaching process were particularly two factors: Adequately illustrating the intervention approaches from the point of view of the experts from the respective subprojects and representing all key elements relevant for conducting equity impact assessments from the point of view of the EQUAL research team.

## 3. Results

The developed logic model framework comprises a general equity-focused logic model (Figure 1) to be adapted to specific interventions resulting in intervention-specific equity-focused logic models (Figure 2, Figure 3, Figure 4, Figure 5 and Figure 6) for equity impact assessments. The following sections provide detailed descriptions of the general equity-focused logic model and how to adapt it to specific interventions for equity impact assessments, using the AEQUIPA interventions as practical examples.

### 3.1. General Equity-Focused Logic Model

The general equity-focused logic model consists of nine key elements relevant for equity impact assessments of interventions. The nine key elements are represented by nine boxes arranged in four rows. The lowest row comprises the element *Theory/conceptual approach*, which provides the basis for any intervention. The second row comprises the element *Personnel and financial resources*, which are used to turn theories or conceptual approaches into practice. The third row consists of three elements representing the common stages of the intervention process: *Recruitment*, *Intervention*, and *Outcome evaluation*. The fourth row shows three elements representing the three key issues contributing to an intervention’s overall impact on equity: *Study participation*, *Compliance with and acceptance of intervention*, as well as *Efficacy of intervention*. Each of these three elements includes a symbol of an ‘equity lens’ representing the following three questions to be answered when conducting equity impact assessments of interventions:Are there social inequalities in study participation?Are there social inequalities in compliance with and acceptance of the intervention?Are there social inequalities in intervention efficacy?

These eight elements are connected with arrows representing causal (thin arrows pointing up) or temporal relationships (thick arrows pointing to the right). The six elements in the upper two rows are enclosed by a ninth element *Intervention level* representing the main level at which an intervention is conceptualized (i.e., individual, contextual, policy).

### 3.2. Adapting the General Equity-Focused Logic Model to Specific Interventions for Equity Impact Assessments

Developing intervention-specific equity-focused logic models implies taking the general logic model as a template and adapting it to the specific characteristics of the respective interventions. We recommend starting the adaptation process with defining the *Intervention level*, followed by specifying the boxes on *Theory/conceptual approach*, *Personnel and financial resources*, *Recruitment*, *Intervention*, and *Outcome evaluation*. The three boxes on *Study participation*, *Compliance with and acceptance of intervention*, as well as *Efficacy of intervention* and associated equity lenses require no further specification, since these elements include the three key questions to be answered when conducting equity impact assessments. During the adaptation process, the structure of the logic model can be modified by adding, refining, or removing single elements, if needed.

#### 3.2.1. Intervention Level 

The *Intervention level* element indicates the main level at which an intervention is conceptualized and implemented, distinguishing between the individual, contextual, and policy level. Interventions at the individual level address directly the people in need of increasing their PA behavior. Interventions at the contextual level aim to modify the social and built environmental factors relevant for being physically active. Interventions at the policy level address the legal and institutional circumstances and (inter-)sectoral policies shaping the contextual level. Interventions may also be conceptualized and implemented on multiple levels (‘multilevel interventions’).

Within AEQUIPA, the interventions in PROMOTE (Figure 2) and TECHNOLOGY (Figure 3) both aim to promote PA at the individual level, whereas the intervention in RTC (Figure 5) acts at the contextual level. Since the interventions in OUTDOOR ACTIVE (Figure 4) and AFOOT (Figure 6) are implemented on multiple intervention levels, additional boxes were added to illustrate the respective multilevel approach.

#### 3.2.2. Theory/Conceptual Approach

The *Theory/conceptual approach* element describes the interventions’ underlying assumptions about how to promote PA behavior. This could be information on certain theories or models of health behavior [55,56], as well as planning models which may guide the choice and systematic application of health behavior theories and conceptual approaches in the planning of interventions [57].

For example, the logic model for PROMOTE (Figure 2) indicates that the intervention is based on the health action process approach (HAPA) [35] and self-regulation theory [36], both focusing on psychosocial processes of behavior change (e.g., self-efficacy, intentions). The intervention in AFOOT (Figure 6) is built on an adapted version of the ecological model of four domains of active living [50], emphasizing the impact of the policy environment on contextual determinants of active transport. The intervention program developed and implemented in OUTDOOR ACTIVE (Figure 4) is based on the PRECEDE–PROCEED model [42] and follows the principles of ecological systems theory [43].

#### 3.2.3. Personnel and Financial Resources

The *Personnel and financial resources* element describes the interventions’ available resources, including personnel, such as staff, volunteers, or cooperation partners, as well as financial, such as research grants or donations. Due to data protection, the personnel and financial resources of the AEQUIPA interventions are not shown in Figure 2, Figure 3, Figure 4, Figure 5 and Figure 6.

#### 3.2.4. Recruitment

The *Recruitment* element includes information on the interventions’ target populations as well as strategies on how these are approached for participation. In PROMOTE (Figure 2), the target population comprises older adults required to meet defined criteria, such as being resident in a selected community, aged 65 to 75 years, able to live independently, as well as having internet access. Random samples of residents’ registration offices were drawn, and potential participants were invited via direct mailing. Additionally, the intervention was advertised via local newspaper articles and public events. In AFOOT (Figure 6), stakeholders in urban planning and public health authorities, policy makers, local stakeholders, and representatives of older adults comprise the target population for participating in the intervention development and implementation process including interviews, workshops, and role-playing games. All stakeholders in urban planning and public health authorities of the study region were invited to participate. Contacts of the participating stakeholders in urban planning and public health authorities were used to identify policy makers, local stakeholders, and representatives of older adults in the respective community.

#### 3.2.5. Intervention

The *Intervention* element indicates the ways in which interventions are designed and delivered. Interventions to promote PA may consist of one or multiple components, provided through one or multiple modes of delivery. The intervention implemented in PROMOTE (Figure 2) was developed according to the recommendations by the World Health Organization [58] and the American College of Sports Medicine [59]. It was delivered at participants’ homes as well as within the community, comprising individually tailored brochures with PA recommendations, web-based diaries, activity trackers, a web-based forum, as well as weekly group meetings. To take account of the participatory components of the interventions implemented in TECHNOLOGY (Figure 3), OUTDOOR ACTIVE (Figure 4), and AFOOT (Figure 6), the boxes on *Intervention* were refined by adding further elements. In AFOOT (Figure 6), the box on *Intervention* comprises four consecutive elements: (1) The development of the guidelines for intersectoral policy action based on findings of a literature search as well as interviews and workshops with local stakeholders, (2) a simulation of the implementation of the guidelines by means of role-playing games, (3) a revision of the guidelines to increase user-friendliness, as well as (4) their comprehensive and in-depth implementation.

#### 3.2.6. Outcome Evaluation

The *Outcome evaluation* element indicates which and how intervention effects are measured. This may include changes in individual PA behavior and related health outcomes, as well as changes in individual or contextual predictors of PA behavior. The intervention implemented in PROMOTE (Figure 2) is evaluated for its impact on changes in objectively and subjectively measured PA level, determinants of PA, such as intentions and self-efficacy, as well as indicators of healthy ageing, such as physical fitness and quality of life. The intervention in RTC (Figure 5) is evaluated for its impact on changes in Community Readiness scores representing the communities’ capacities for PA. Since RTC comprises two rounds of outcome evaluation for assessing the effects of two rounds of intervention, two *Outcome evaluation* elements are included in the model. In AFOOT (Figure 6), the *Outcome evaluation* element includes the assessment of the effects of the comprehensive implementation by means of awareness among local administrative and policy actors, as well as changes in administrative routines. Furthermore, the element includes the evaluation of the in-depth implementation focusing on established capacities and actionable knowledge, as well as changes in administrative routines, public space, and mobility behavior among the elderly population living in a community defined as urban transition lab.

#### 3.2.7. Study Participation

The *Study participation* element and associated equity lens represent the first question to be answered when conducting equity impact assessments of interventions: *Are there social inequalities in study participation?* Given that socioeconomically disadvantaged individuals are less likely to be physically active [7,8,9,10] and socioeconomically disadvantaged contexts often have fewer opportunities for and more barriers to PA [27,60], not reaching these individuals or contexts may result in interventions being ineffective for those that need them the most, thereby further increasing social inequalities. Assessing social inequalities in study participation, therefore, implies evaluating an intervention’s success in reaching individuals or contexts with the greatest need for interventions. Regarding the intervention implemented in PROMOTE (Figure 2), potential social inequalities in study participation are evaluated by stratifying response rates by sociodemographic and socioeconomic characteristics. In OUTDOOR ACTIVE (Figure 4), potential social inequalities in study participation can be evaluated at several time points to identify whether socioeconomically disadvantaged individuals participate in the continuous process of intervention development and implementation. In AFOOT (Figure 6), potential social inequalities in study participation are assessed by evaluating whether participating policy makers and stakeholders are already aware of the relevance of health equity issues.

Once social inequalities in study participation are detected, these may be particularly attributable to the choice of recruitment strategies and eligibility criteria for selecting potential participants. To minimize the risk of inducing social inequalities in study participation, ‘active’ recruitment strategies (i.e., direct interaction with potential participants) and formative research (e.g., focus groups), for example, are assumed to be particularly successful [61,62]. Social inequalities-sensitive recruitment strategies, however, require additional personnel and financial resources, which need to be considered right at the stage of project proposal.

#### 3.2.8. Compliance with and Acceptance of Intervention

The *Compliance with and acceptance of intervention* element and associated equity lens represent the second question to be answered when conducting equity impact assessments of interventions: *Are there social inequalities in compliance with and acceptance of the intervention?* Assessing these inequalities implies evaluating whether the compliance with and acceptance of an intervention differ between individuals or contexts of different SES. Interventions that are less complied with and accepted among socioeconomically disadvantaged individuals or contexts may result in interventions being ineffective for those that need them the most, thereby further increasing social inequalities. Assessing potential inequalities in compliance with and acceptance of the intervention implemented in PROMOTE (Figure 2) includes, for example, the investigation of whether completion of web-based PA diaries or attendance of group meetings differ by sociodemographic and socioeconomic characteristics of individuals. In AFOOT (Figure 6), potential social inequalities in compliance with and acceptance of the intervention are evaluated at several time points of the intervention development and implementation process. For example, the simulation of the implementation of the guidelines by means of role-playing games is evaluated with regard to whether aspects of health equity and environmental justice [63] are considered by stakeholders in urban planning and public health authorities, policy makers, and local stakeholders.

If social inequalities in compliance with and acceptance of interventions have been detected, these may primarily be due to the ways in which interventions are developed, designed, and delivered. To minimize the risk of inequalities in compliance with and acceptance of individual- and contextual-level interventions, participatory approaches, for example, are assumed to be particularly successful. Actively involving socioeconomically disadvantaged individuals or contexts in the intervention development and implementation process can help to better tailor interventions to the specific preferences and needs of these individuals or contexts. For policy-level interventions, using a win-win approach demonstrating the co-benefits of intersectoral collaboration may foster the consideration of aspects of health equity and environmental justice among stakeholders and policy makers [64].

As with social inequalities-sensitive recruitment strategies, social inequalities-sensitive intervention strategies also require additional personnel and financial resources, which should be considered right at the stage of project proposal.

#### 3.2.9. Efficacy of Intervention

The *Efficacy of intervention* element and associated equity lens represent the third question to be answered when conducting equity impact assessments of interventions: *Are there social inequalities in intervention efficacy?* Assessing social inequalities in intervention efficacy implies analyzing whether intervention effects differ between individuals or contexts of different SES. Interventions being less effective or even ineffective among lower SES individuals or contexts may further increase social inequalities, both at the individual and contextual level. For the intervention implemented in PROMOTE (Figure 2), equity-specific subgroup analyses [65] are conducted to assess whether intervention effects differ between individuals of different sociodemographic and socioeconomic characteristics. Given that measuring effects on inequalities in relative (ratio) or absolute (difference) terms can lead to divergent results and conclusions regarding whether interventions reduce, increase, or have no effects on inequalities [66,67], both relative and absolute measures of inequalities are calculated. In AFOOT (Figure 6), after the comprehensive and in-depth implementation, potential social inequalities in intervention efficacy are assessed from the perspective of ‘Health Equity in All Policies’ [68] by evaluating whether and to what extent aspects of health equity and environmental justice are integrated in administrative routines, intersectoral collaborations, and planning processes.

Once social inequalities in intervention efficacy are detected, these may be attributable to the ways in which an intervention is conceptualized, developed, and delivered, based on the underlying theoretical or conceptual assumptions about how to promote PA behavior. For example, studies have shown that theories of behavior change may work differently in individuals of different SES [69,70,71]. To minimize the risk of inducing social inequalities in intervention efficacy, interventions should be built on theories or models that already consider social inequalities. Ecological models refer to aspects of social inequalities, e.g., by integrating concepts of environmental justice [63]. Concepts of public administration, such as ‘joining up’, ‘boundary crossing’, and ‘partnerships’ [72], theories of policy process with interactions between political, administrative, private, community, and expert actors [73], as well as complex adaptive system thinking [74] are further examples relevant for public health approaches of intersectoral actions to tackle social determinants of health.

## 4. Discussion

Reducing social inequalities in health is a priority for public health. Despite the plea to evaluate all public health interventions for their impact on inequalities [18], the evidence regarding equity impacts of interventions to promote PA is limited [20,21,22,23,24,25,26,27]. We proposed a logic model framework for equity impact assessments of interventions to promote PA among older adults. The framework includes a general equity-focused logic model to be adapted to specific interventions resulting in intervention-specific equity-focused logic models. These models comprise the key elements relevant for assessing social inequalities in study participation, compliance with and acceptance of interventions, as well as intervention efficacy.

Using the proposed logic model framework has multiple strengths. First, using the framework for developing equity-focused logic models helps researchers to systematically apply an equity lens to the evaluation of PA interventions. By laying out an intervention’s key elements relevant to equity and the relationships between them, equity-focused logic models also help planners, implementors, and evaluators of interventions to better understand whether, how, and why an intervention affects equity, and thus allow to derive implications for tackling social inequalities in future interventions. Moreover, although developed in the area of PA promotion for older adults, the framework can be applied to conduct equity impact assessments of public health interventions focusing on other health outcomes and populations. Finally, equity-focused logic models may not only be used during intervention evaluation, but also during the planning and implementation stage of interventions. Planning equity-specific analyses right at the beginning of a research process is even considered to result in more reliable evidence on equity impacts [65]. In this regard, using our logic model framework already at the planning stage of interventions can help to consider the collection of data relevant for analyzing equity impacts.

There are, however, certain challenges to using our logic model framework to be considered. In general, a main challenge when developing logic models is to strike an adequate balance between simplicity and complexity [75]. Equity-focused logic models of interventions should focus on the interventions’ key elements relevant for equity impact assessments, retaining enough detail to not oversimplify the complex relationships between them. Furthermore, the use of logic models generally requires flexibility in reacting to potential changes [75]. For instance, equity-focused logic models may need revision to consider equity-relevant issues arising within the course of the research process. Finally, developing equity-focused logic models as well as interventions suitable for equity impact assessment requires additional temporal, personnel, and financial resources, requiring adaptations of the entire research process. These resources must be considered when applying for research grants. 

A strength of this study is the inclusion of multiple sources of evidence, which has proven to be beneficial for developing logic models [76]. We realized interviews with experts covering all areas relevant for filling in the knowledge gaps identified in our systematic review. Moreover, in line with general recommendations on developing logic models [31], the general and intervention-specific equity-focused logic models were developed in collaboration with multiple stakeholders involved in the development, implementation, and evaluation of PA interventions. At this time, a limitation of our study is that, except for presentations at one national and one international scientific conference [77,78], no external feedback from potential users of our framework outside of our own research network has been sought. Moreover, concrete details on the practical realization of equity impact assessments are not yet included. To overcome these limitations, in the second funding phase of the prevention research network AEQUIPA, external feedback from potential users of our framework will be sought and the framework will be applied to conduct equity impact assessments of the AEQUIPA interventions. Furthermore, as part of an international cooperation on a joint equity-specific analysis of intervention effects, our logic model framework will be applied to other European PA interventions conducted beyond our research network. Based on our experiences within AEQUIPA and feedback received from the cooperating research teams, we will revise the general equity-focused logic model and related explanations, which will further improve the usability of our framework. 

## 5. Conclusions

Reducing social inequalities in health is a major challenge for public health. The proposed logic model framework can be used to conduct equity impact assessments of interventions, which is a major prerequisite for the prioritization of interventions most likely to be effective in reducing health inequalities. Future work within the AEQUIPA prevention research network on the equity impacts of the implemented interventions at individual-, contextual-, or policy-level will reveal which elements of the intervention design and implementation are most critical for the interventions’ impact on social inequalities in PA.

## Figures and Tables

**Figure 1 ijerph-16-00420-f001:**
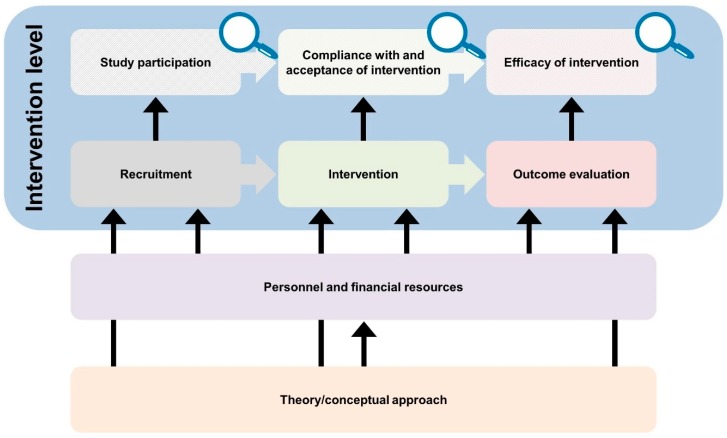
General equity-focused logic model.

**Figure 2 ijerph-16-00420-f002:**
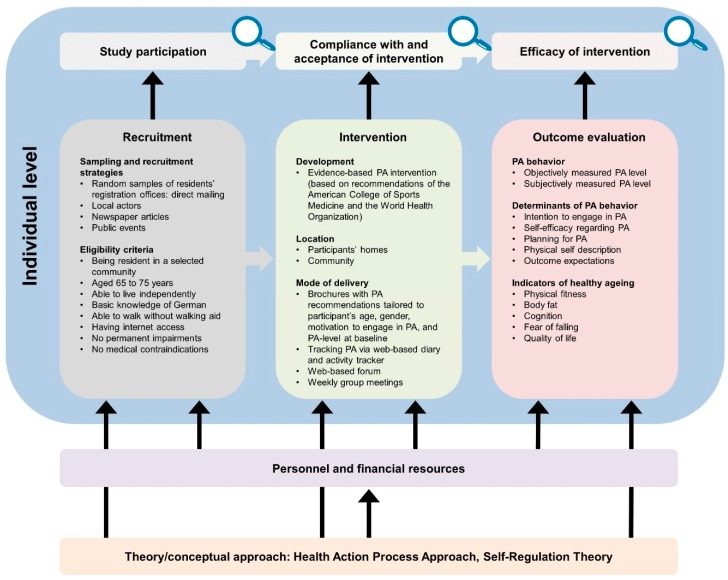
Specific equity-focused logic model for the individual-level web-based tailored intervention aimed at promoting physical activity (PA) and fitness in older adults implemented in PROMOTE.

**Figure 3 ijerph-16-00420-f003:**
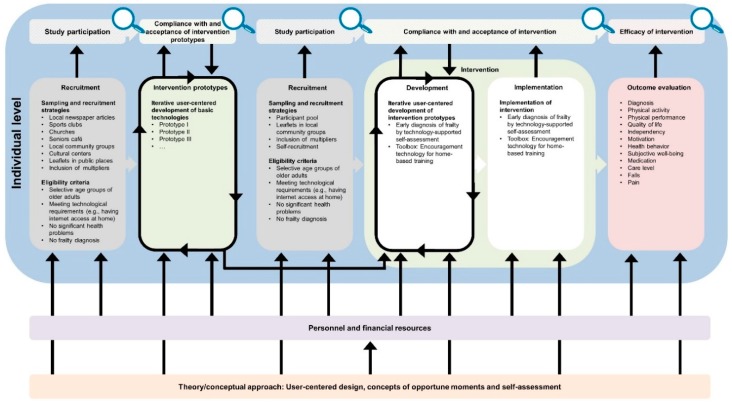
Specific equity-focused logic model for the individual-level technology-based individualized intervention aimed at encouraging physical activity and preventing functional decline in older adults implemented in TECHNOLOGY.

**Figure 4 ijerph-16-00420-f004:**
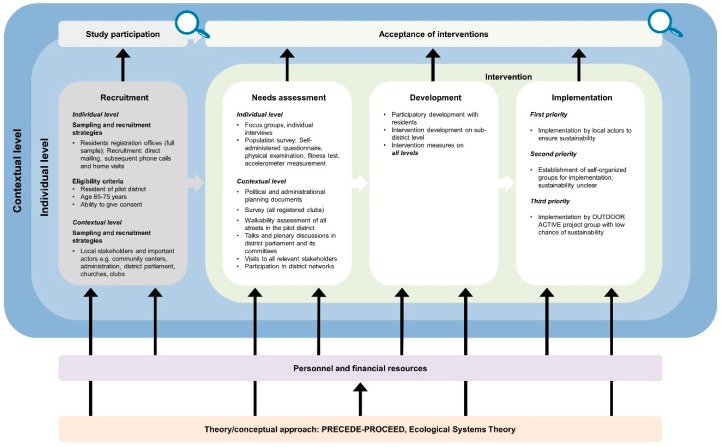
Specific equity-focused logic model for the multilevel participatory community-based outdoor physical activity program for older adults implemented in OUTDOOR ACTIVE. In funding phase one, OUTDOOR ACTIVE was designed as a pilot study and did not contain an outcome evaluation. In funding phase two, the efficacy of the developed intervention approach will be evaluated in a cluster-randomized trial [44].

**Figure 5 ijerph-16-00420-f005:**
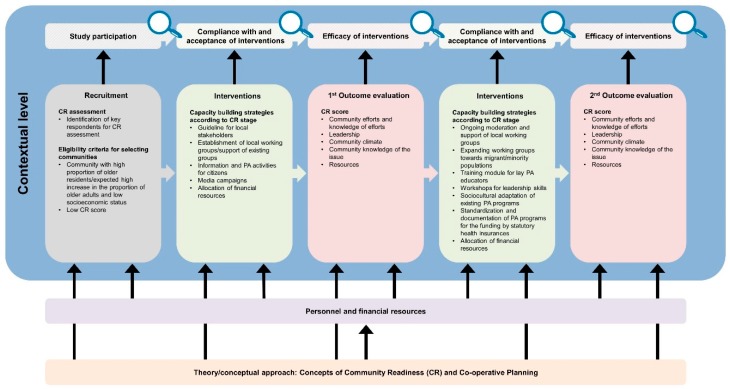
Specific equity-focused logic model for the contextual-level intervention to increase community readiness (CR) to engage older adults in physical activity (PA) interventions implemented in RTC.

**Figure 6 ijerph-16-00420-f006:**
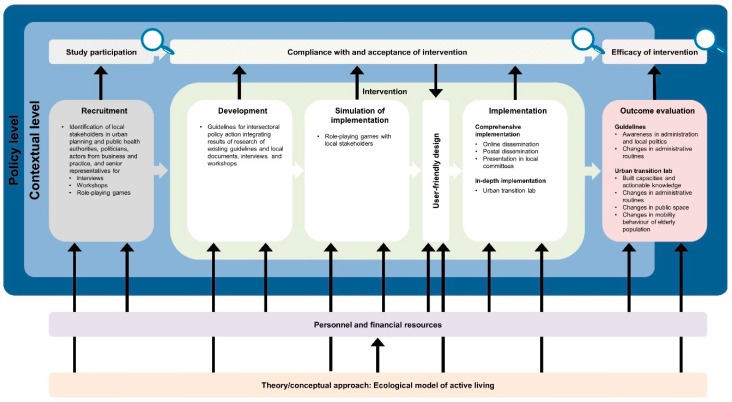
Specific equity-focused logic model for the contextual- and policy-level intervention focusing on intersectoral policy action to promote mobility and age-friendly municipal development implemented in AFOOT.

**Table 1 ijerph-16-00420-t001:** The five physical activity interventions implemented within AEQUIPA.

AEQUIPA Subproject	Intervention Approach
*Individual-level interventions*	
(1) PROMOTE	Web-based tailored intervention aimed at promoting self-monitoring of physical activity and fitness in older adults based on behavior change techniques (goals and planning, feedback and monitoring, social support, shaping knowledge, comparison of behavior [34]) according to the health action process approach [35] and self-regulation theory [36,37,38,39]
(2) TECHNOLOGY	Technology-based individualized intervention aimed at encouraging physical activity and preventing functional decline in older adults by providing environmental prompts [34] in opportune moments ^1^ via diverse unobtrusive, portable, ubiquitous sensing and feedback technologies, as well as self-assessments of physical performance [40,41]
*Individual- and contextual-level intervention*	
(3) OUTDOOR ACTIVE	Participatory community-based outdoor physical activity program for older adults based on the PRECEDE–PROCEED model [42], including the participatory development and implementation of intervention measures aimed at promoting outdoor physical activity following the principles of ecological systems theory [43,44]
*Contextual-level intervention*	
(4) RTC	Strategies to increase community readiness (community knowledge of the issue, community knowledge of efforts, community climate, leadership, resources [45,46,47]) to engage older adults in physical activity interventions [48,49]
*Contextual- and policy-level intervention*	
(5) AFOOT	Guidelines for actors in municipal administrations aimed at fostering intersectoral policy action to promote mobility and age-friendly municipal development according to an adapted version of the ecological model of four domains of active living [50,51,52]

^1^ Opportune moments are defined as points in time when an individual has the capability and willingness to get reminded about and conduct physical activities.

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
