# Peer review of "Equity Impact Assessment of Interventions to Promote Physical Activity among Older Adults: A Logic Model Framework"

_ijerph, 2019, doi:10.3390/ijerph16030420_

Reviewer 1 Report

I am appreciative of this manuscript. It investigates an under researched topic and provides some good practical directions for researchers. The manuscript is well written and fluid and I would consider this one of the best papers i have reviewed this year.

There is only one major issue which the authors do need to tackle as it is central to their whole paper. The authors provide 5 figures of the different logic models but really don’t unpick these so they are useful for readers, particularly I am thinking of individuals who may be less familiar with those.   It makes it difficult for readers to fully understand how the logic models are used in practice.  The narrative provided also does not explicitly link to the figures. It would be most useful if the authors could provide a narrative walk through of at least one of these figures to better guide readers and link the existing narrative to the logic models.

At the moment it feels a little disconnected in terms of the logic models themselves and what is written. So in the discussion (see end P28) it states ‘Using the proposed logic model…’ but it isn’t clear how readers use actually use the logic model.  As with the aforementioned suggestion, it would be beneficial to expand a little, almost in a ‘how to’ way to guide people who want to use this approach.

Author Response

Please find attached our point-by-point response to the comments of reviewer 1.

Reviewer 2 Report

Reviewer comments

Thanks for giving me the opportunity to read through this paper. Authors have made efforts to develop a logic model to help other researchers who may wish to conduct equity impact assessments of interventions that aim to enhance physical activity. While the aims of this work are praiseworthy, I have some concerns about the completeness of the logic model development process and the usefulness of the findings. Specifically, at present, authors provide too few methodological details and the model that they propose requires clarification and further explanation, ideally with clear examples provided. Once these details are added, this paper may be able to be accepted as potentially suitable for publication.

Abstract

Line 27 = ‘The aim’

Line 30 = please define what ‘AEQUIPA’ means in the abstract

Line 34: interventions’ equity impacts

General – could you please explain why researchers might need a logic model framework– the abstract is quite technical and detailed, and you may find that non-expert audiences won’t understand exactly what the purpose of this work is from the content you have given in the abstract.

Introduction

Line 5 – please include a definition of the term ‘inequalities’

Line 22-24 – please define what a ‘logic model’ is at some point in this paragraph (also see general comments above for the abstract)

General – once again, this is a very factual and tight description of the research. To bear in mind, your audience may not be aware of what all the terms that you refer in this section actually mean, so it would help the reader if you can add in a few explanations and/or examples of some of the concepts that you refer to.

Materials and Methods

Table 1 – could you add some extra descriptive information on what each study intervention involves (e.g. ‘Technology-based individualized intervention’ – is this pedometers/an app/ something else??). You could think about summarizing in terms of the Behaviour Change Technique Taxonomy (e.g. Susan Michie’s work at UCL).

Section 2.2 line 5 – could you please provide some details of how you recruited your interviewees and what criteria for selection you used? At 9 participants, this is also a very small qualitative sample. Could you provide justification (in discussion) for why you chose to stop your interviews at 9 people, rather than continue with a wider sample?

Line 9 - content analysis – please can you provide a description of the steps that you took in this content analysis, and how you ensured that you were using the method appropriately. Could you also include in an appendix the list of interview questions that you asked and an explanation or justification for how you developed these?

Line 10 – please provide details of your literature search teams and search approach – see the PRISMA statement for guidance on how best to report details of this. It would also be useful t see a flow diagram in the results section. The paper also needs a rationale for why you took the search approach that you did, and why you chose the specific search terms that you did.

Line 16 - the sentence ‘The draft of the general logic model was transferred to the interventions implemented in the AEQUIPA subprojects’ – what exactly do you mean by the term ‘transferred’ – should this read as ‘piloted’?

Line 17 – please could you provide further information of the steps that you took during your ‘iterative process’ – specifically, how many rounds of iteration, on what criteria did you choose to make any changes, how many people were involved and what was their expertise in the area?

Results

Page 3, Line 24 – please define  the meaning of ‘transferred’

Top paragraph on page 4 – lines 1 to 20 – this is very hard to understand and I suggest you consider rewriting it. From the description given, I can’t unfortunately understand how to use the diagram you provide in figure 1. My recommendation is you try to re-write this and either re-write or remove the technical terminology, and then check with an audience who are non-expert to see if they can understand how to use your tool.

3.2. all sections – please provide an example of how to use each element in the model, based on the studies you mention in figure 1, in the text. So, you walk the reader through the exact process you would like them to use if they adopt your logic model with clear examples. This would help clarity and readability. Where you provide examples in figs 2 – 4, you need to explain the implications of the points that you highlight in text for equity.

General  - I recommend that this section is re-structured and re-written so that it is much easier to read and the benefits of using the logic model are clearly identified and explained to the reader.

Discussion

Please provide commentary on when in the study planning and execution process this logic model should be used.

My main thought is that the proposed framework needs to be better explained in order to overcome your stated limitation of ‘strik[ing] the adequate balance between simplicity and complexity’. Have you asked others to use this model (e.g. outside of our research group) and adapted the tool based on feedback as to whether they find it easily useable/ helpful to their work? You may want to consider re-working the model based on this type of feedback to improve it somewhat.

Please add in strengths and limitations of your chosen research approach and how this could be improved. It would also be useful to add in some information on how others have gone about developing logic models, and whether you consider your process to be better/worse/different and how?

Author Response

Please find attached our point-by-point response to the comments of reviewer 2.

Round  2

Reviewer 2 Report

Thanks for taking time to address the comments in my first review, the paper is reading a lot more clearly and fleshing out the methods section is really useful for the reader to understand the steps you have taken.